environmental science/environmental engineering/energy

Chinese solar greenhouse, ridge position ratio, thermal environment, numerical simulation, structure design

**Author for correspondence:**
Yiming Li
e-mail: liyiming@syau.edu.cn

# Effect of the ridge position ratio on the thermal environment of the Chinese solar greenhouse

Xiaoyang Wu[1,3], Xingan Liu[1,3], Xiang Yue[2,3], Hui Xu[1,3], Tianlai Li[1,3] and Yiming Li[2,3]

[1]College of Horticulture, and [2]College of Engineering, Shenyang Agricultural University, No. 120 Dongling Road, Shenhe District, Shenyang 110866, People's Republic of China
[3]National and Local Joint Engineering Research Center of Northern Horticultural Facilities Design and Application Technology (Liaoning), No. 120 Dongling Road, Shenhe District, Shenyang 110866, People's Republic of China

YL, 0000-0001-8764-5589

This paper clarified the mechanism of the south and north roofs' effect on the thermal environment of the Chinese solar greenhouse (CSG), using a new parameter: ridge position ratio (RPR), which can describe the dynamic dependency relationship between the south and north roofs. A mathematical model was established using a method of combining computational fluid dynamics (CFD) simulation with experiments, then the model was used to further analyse the effect of RPR on the thermal environment of the CSG. The experimental greenhouse was simulated as an empty building to obtain results independently from these factors including crop and ventilation conditions. The results showed that the occurrence time of the maximum air temperature will be delayed when RPR increases to 0.3 during the daytime. As RPR increases, the heat storage layer of the soil gradually becomes thinner, but the north wall remains unchanged. RPR has a relatively small effect on the minimum temperature of each greenhouse part during the night. Mathematical models of the relationships between RPR, the solar energy that entered the greenhouse and the released heat energy of the enclosure structures were established, respectively. This paper can provide theoretical guidance for the structural design of the CSG.

## 1. Introduction

By the end of 2018, the area of the solar greenhouses had reached 577 000 hectares in China [1]. The Chinese solar greenhouse (CSG) not only completely solved the supply problem of vegetables in

the north region (42.5° N region) during the winter, realizing the goal of annual vegetable production, but also fully used solar energy without any auxiliary heating, which saved a lot of energy and achieved remarkable economic and social benefits [2,3]. With the potential to promote agricultural modernization and augment the growth of horticulture in China, CSG can make a significant contribution to sustainable development [4]. CSG consists of a north wall, gables, north and south roofs. The south roof is covered with a thin plastic film during the daytime and a heat preservation quilt at night. The growth of the crop depends on the balance between various environmental factors, including temperature, solar radiation, carbon dioxide concentration, relative humidity, and so on [5,6]. In particular, the thermal environment of the greenhouse is significantly important for crops growth [7]. It is necessary to create a reasonable light and thermal environment, which are the key factors that play an important role in the growth and development of the crops. An inappropriate structure would lead to undesired temperature regulations inside the greenhouse and fail to provide the optimal conditions for crops growth [8,9]. The shape coefficient of the solar greenhouse is large, and the geometric size directly affects the thermal environment. Therefore, the efficiency of the solar greenhouse can be improved by selecting reasonable space to form key parameter values. The commonly used building space parameters of the solar greenhouse mainly include span, height of the roof ridge, horizontal projection length of the north roof and height of the north wall. The typical design of the solar greenhouse uses an exorbitant ridge, resulting in oversized space under the roof. It should be noted that the horizontal position of the ridge of the solar greenhouse determines the area ratio of the south roof and north roof and then affects the thermal environment of the solar greenhouse.

The size of the south roof not only determines the amount of solar energy received by the greenhouse during the daytime, but also directly relates to the amount of heat released to the outside at night. The south roof is covered with the heat preservation quilt at night, but its thermal resistance is still much lower than the north wall [10]. It is estimated that the heat loss through the south roof accounts for 70–80% of the total heat loss in the greenhouse. The increased area of the south roof allows more solar energy to enter the greenhouse during the day, but also causes more heat to be lost at night [1]. The north roof is a non-transparent envelope structure. Compared with the north wall, the north roof is mainly used for heat preservation due to the sunshine duration on the inner surface of the north roof being limited, and its heat storage is very limited. If the area of the north roof is excessive, it will affect the capacity of the south roof to intercept solar energy, which is not conducive to crop photosynthesis, while if the area is too small, it will lead to excessive heat loss at night. Therefore, the reasonable determination of the area of the south and north roofs is essential to balance the solar energy that enters the greenhouse and reduce the greenhouse heating at night. It is worth paying attention to if the horizontal position of the ridge moves to the north, the area of the south roof increases, and the area of the north roof decreases. On the contrary, if the horizontal position of the ridge moves to the south, the area of the south roof decreases, and the area of the north roof increases. Therefore, it is essential to study the effect of different horizontal positions of the ridge on the thermal environment of the solar greenhouse, which makes the amount of solar energy that enters the greenhouse during the daytime and the heat that is released from the greenhouse during the night optimized.

To date, most researchers have studied mainly the morphology parameters of the south roof and the north roof on the thermal environment of the solar greenhouse. The amount of solar energy entered the greenhouse can be increased by raising the area of the south roof, but the increase in energy is limited because the heat released from the enlarged south roof at night is greater [11]. The projected width of the north roof and the height of the north wall are based on the variation of the projection height of direct sunlight on the north wall on different dates [12]. Due to the difficulties and limitations of experiments, most of the research on the south roof adopts numerical models. The different shapes of the south roof on the thermal environment of the solar greenhouse were analysed by comparing the instantaneous light transmittance and the solar flux to each surface of the greenhouse [13–16]. In addition, by comparing the experimental data, the height and span ratios of different values obtained in the solar greenhouse range from 0.51 to 0.55. There is an optimal value for the height and span ratio, which provides the highest solar energy collection and raises the internal air temperature during the winter [17]. Through the analysis of the above research results, the relative dimension of the south roof and the north roof has an obvious effect on the thermal environment of the solar greenhouse. However, in the published literature, there is insufficient information about the comprehensive analysis of the south and north roofs combination, and no parameter coupling the south and north roofs.

The present research used a new parameter: ridge position ratio (RPR), which can describe the dynamic dependency relationship between the south and north roofs, and then clarified the mechanism of the south and north roofs' effect on the thermal environment of CSG. Previous experiments were

conducted on site. Some differences may have existed, such as different crop conditions, different wall heights or thicknesses, different materials and so on. Moreover, traditional experimental studies are unrepeatable, geographically restricted and it is difficult to obtain all information. Thus, more studies are needed to better understand how to select appropriate RPR based on comparisons of the solar greenhouses with identical conditions. Some thermal models have been developed for the prediction of temperature variations of different interactive components in CSGs including inside air, soil, north wall and gables [18–22]. The simulation model CSGHEAT has been developed based on the energy balance equation to estimate the hourly heating requirements in a CSG [20]. With the set indoor temperatures, the surface temperatures of the soil and north wall were estimated by solving ordinary differential heat balance equations. Computational fluid dynamics (CFD) is an effective method of numerical analysis and research on fluid flow and heat conduction through computer numerical simulation and visual processing. CFD can calculate accurately the distribution of solar energy in the greenhouse compared with the model based on the energy balance equation, instead of assuming that the solar radiation available to walls and soil is evenly distributed. In the past 30 years, CFD has been mainly applied in the simulation and optimization design of greenhouse structure and configuration to achieve the purpose of energy saving [23–29]. Tong *et al.* [25,26] have used CFD for simulation of time- and space-dependent temperature distributions in a CSG, then three groups of span configurations (10, 12 and 14 m) were analysed, using CFD simulation for span dimension selection of CSG and their characteristics including solar heat gains, heat losses and temperature distributions. Zhang *et al.* [27] used CFD simulation to determine the best thickness of the north wall and employed an evaluation model using weighted entropy and fuzzy optimization for decision-making. He *et al.* [28] analysed how the dimensions of the north wall vent influenced CSG temperature control using the CFD method. Chen *et al.* [29] examined the effect of the water-cycle heating system on a greenhouse and used CFD numerical simulation to predict changes in the environment and the energy consumption in greenhouses. Most importantly, all input parameters in the CFD model can be easily controlled, allowing them to be constant throughout the simulation, while uncertainty is inevitable in experimental measurements.

This study established a mathematical model by using the CFD numerical simulation method, which is based on the thermal theory of buildings and considers the structural characteristics of the solar greenhouse. Compared with the simulation model CSGHEAT, the model in this study considers the distribution of solar energy in the greenhouse and energy transfer and can accurately calculate three-dimensional heat flux distributions for all the enclosure structures including walls and soil by calculating partial differential equations, which makes the calculation results more realistic. Once the model had been validated, it was applied to analyse the influence law of different RPR on the thermal environment of CSG. Nine RPR (from 0.10 to 0.45) models were established and all the models used the same boundary and initial conditions. First, we compared temperature distributions of CSGs with different RPR based on this model, including the maximum and minimum temperatures of air and solid surfaces inside the greenhouse, the interior temperature distribution of the soil and the north wall. Second, the effects of RPR on the amount of solar energy that entered the greenhouse were studied using statistical analysis on the model simulation results. Finally, the effects of RPR on heat-released capacity of the envelopes of the greenhouse at different moments were analysed.

# 2. Material and methods

Numerical simulation and experimental verification were carried out in the institute of Horticultural Facility Design and Environmental Control, Shenyang Agricultural University (41.8° N, 123.4° E, 42 m altitude height). There was no crop in the greenhouse, and ventilation and humidity exchange conditions in the greenhouse were not considered. The experimental greenhouse was simulated as an empty building to obtain results independent of these factors. Figure 1 shows the experimental greenhouse consisting of the north wall, the north roof, the south roof and the soil below the greenhouse. The greenhouse has a span of 10 m, a length of 60 m and a ridge height of 6 m. It faces south and has an azimuth of 7° south by west. The north wall, which consists of a 0.37 m fly ash brick inside and a 0.11 m polystyrene board outside, has a height of 3.8 m. A thin plastic film was used to cover the south roof during the daytime (8.30–16.00), allowing sunlight to enter and be absorbed by the greenhouse. A 0.03 m thick heat preservation quilt was used to cover the south roof during the night (16.00–8.30 day+1), preserving the heat mass inside the greenhouse. The experimental period was 1 to 28 January 2017, a typical severe winter period, and the typical sunny day on 23 January was selected as the experimental comparison in this paper.

(a)                                              (b)

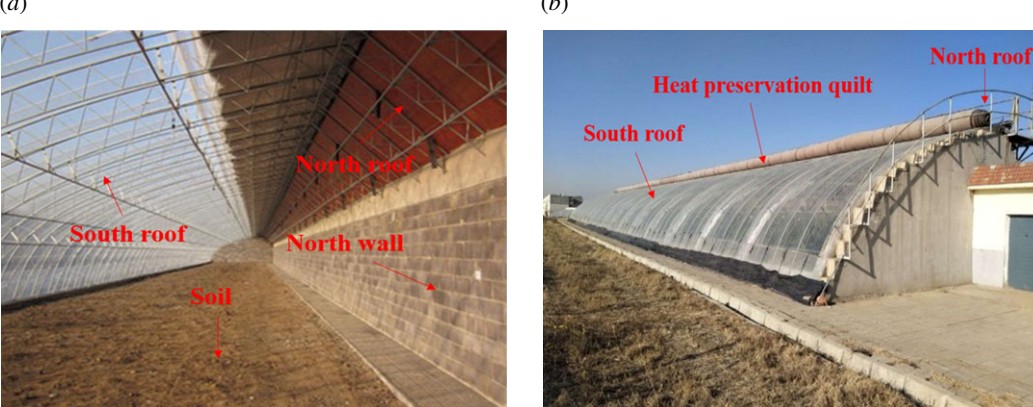

**Figure 1.** The exterior and interior of the experimental CSG.

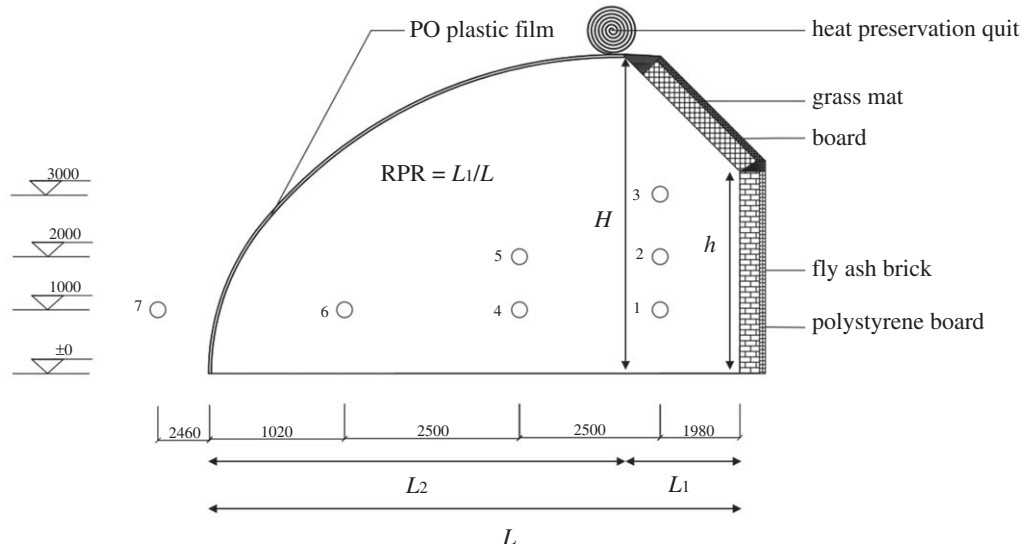

**Figure 2.** Temperature measure points inside and outside the experimental CSG.

The thermal environment inside the CSG was monitored for 24 consecutive hours to obtain an integrated evolution cycle in a single day (i.e. from 0.00 to 0.00 the next day). The measuring point of indoor and outdoor air temperature was 1 through 7. In order to effectively analyse the greenhouse microclimate, a TYD-ZS2 environmental data recorder was installed outdoors 1.0 m away from the greenhouse to monitor the external temperature of the greenhouse and provide environmental parameters for the numerical model. Figure 2 shows the six measurement locations in the greenhouse to obtain air temperature data in the greenhouse. At each experimental point, the temperature was measured every 1 min and in the meantime was recorded to obtain the exact temperature [30]. The sensor probes were exposed to the air and were radiation-proof during the measurement, and all sensors were in the middle cross-section of the greenhouse. The HOBO automatic temperature recorder was the instrument used to automatically collect the data and finally output in Excel format.

The transport equations of heat transfer in the greenhouse are based on continuity, momentum and energy equation is governed by the following [31–33]:

$$\frac{\partial \rho}{\partial t} + \nabla \cdot (\rho \vec{V}) = 0, \tag{2.1}$$

$$\frac{D\vec{V}}{Dt} = \vec{f}_b - \frac{1}{\rho}\nabla p + \frac{\mu}{\rho}\nabla^2 \vec{V} + \frac{1}{3}\frac{\mu}{\rho}\nabla(\nabla \cdot \vec{V}) \tag{2.2}$$

and

$$\rho\frac{D}{Dt}\left(\hat{u} + \frac{V^2}{2}\right) = \rho\vec{f}_b \cdot \vec{V} + \nabla \cdot (\vec{V} \cdot \tau_{ij}) + \nabla(\lambda\nabla T) + \rho\dot{q}, \tag{2.3}$$

where $\rho$ is the air density, $t$ is the time, $\vec{V}$ is the velocity vector, $\vec{f}_b$ is the volume force item, $p$ is the pressure, $\mu$ is the dynamic viscosity, $\hat{u}$ is the internal energy of air per unit mass, $V$ is the velocity of air per unit mass, $\tau_{ij}$ is the effective viscosity shear, $\lambda$ is the thermal conductivity, $T$ is the temperature, and $\dot{q}$ is the unit mass of air receives heat from the outside through radiation.

The standard $k - \varepsilon$ model was reported to be satisfactory [34,35], which was successfully applied in similar conditions and inside the environment. For most natural convection problems, the Boussinesq model can be used to obtain a faster convergence rate. The density of the greenhouse air has a little variation, so the approximation is accurate [36–40].

In recent studies, CFD was used to simulate the effects of solar radiation that constantly affects heat balance, which can either boost or reduce plant growth inside a greenhouse. In this paper, the P-1 radiation model (equation (2.4)) was applied to the solar radiation, which is used for the heat transfer in the greenhouse [41].

$$q_r = -\frac{1}{3(a + \delta_s) - C\delta_s}\nabla G, \tag{2.4}$$

where $a$ is the absorption coefficient, $\delta_s$ is the scattering coefficient, $G$ is the incident radiation and $C$ is the linear-anisotropic phase function coefficient. The transport equation for $G$ is

$$-\nabla q_r = aG - 4a\delta T^4. \tag{2.5}$$

The expression of equation (2.5) was directly substituted into equation (2.3) to consider the heat source caused by radiation. The solar ray tracing model (equation (2.6)) was activated, which is based on a ray-tracing algorithm to simulate the dynamic changes of solar radiation.

$$\frac{dG(\vec{r},\vec{s})}{ds} + (a + \sigma_s)G(\vec{r},\vec{s}) = an^2\frac{\sigma T^4}{\pi} + \frac{\sigma_s}{4\pi}\int_0^{4\pi} G(\vec{r},\vec{s}')\Phi(\vec{s}\cdot\vec{s}')d\Omega'. \tag{2.6}$$

The microclimate in the solar greenhouse changes dynamically with time, a transient three-dimensional model was adopted. The boundary conditions and initial conditions were determined according to the actual data measured. The inside and outside air temperatures of the greenhouse were 12°C and −20°C initially. The boundary conditions mainly include north and south roofs, soil, north wall setting inside and outside the experimental greenhouse. The south roof was covered with a thin plastic film from 8.30 to 16.00 and a heat preservation quilt for the rest of the time. Customized outdoor temperatures were applied to north and south roofs, north wall exterior surfaces through user-defined functions. The inside surface temperatures of the north wall, soil, north and south roof were 17°C, 15°C, 12.5°C and 10°C initially. The standard wall function was applied to near-wall processing [42]. The finite volume method was used to solve the discrete flow field, while the semi-implicit method for pressure-linked equations (SIMPLE) algorithm was used to solve conservation equations [43]. The solution settings of the gradient discrete scheme were configured as least-squares cell based. The discrete schemes of pressure, density, momentum and energy were all configured as a second-order upwind scheme. The discrete schemes of $k$ and $\varepsilon$ were configured as the first-order upwind scheme. Relaxation factor of pressure was configured as 0.3, density and body force and turbulent viscosity and energy were all configured as 1, $k$ and $\varepsilon$ were configured as 0.8, and momentum was configured as 0.7. The convergence accuracy of continuity equation, momentum equation, $k$ equation and $\varepsilon$ equation was configured as $1 \times 10^{-3}$, and the convergence accuracy of energy equation and radiation equation was configured as $1 \times 10^{-6}$.

As figure 2 shows, the geometric parameters of the solar greenhouse model mainly included the span of the greenhouse $L$, the height of the north wall $h$, the ridge height $H$, the horizontal projection length of the south roof $L_2$ and the horizontal projection length of the north roof $L_1$. The RPR is the ratio of the horizontal projection length of the north roof $L_1$ to the span $L$ when the ridge height $H$ and the span $L$ are constant. The present study used the experimental greenhouse in Shenyang Agricultural University as the standard model (RPR = 0.22). The north wall was composed of 370 mm fly ash brick and 110 mm polystyrene board. Table 1 lists the nine RPR (0.10 to 0.45) models that were established according to it being necessary to ensure that at least part of the north wall is exposed to direct sunlight from 10.00 to 14.00 [12]. The thermophysical properties of the envelope materials in the solar greenhouse are shown in table 2 [44].

The geometry and meshes for the numerical analysis were designed using the Solidworks and Fluent Meshing software tools. The average air temperature (0.00) inside the greenhouse was monitored and checked to reach a steady number. As can be seen in figure 3, the initial number of cells was about

**Table 1.** Geometric parameters of nine RPR models.

| geometric parameter | | | | | |
| --- | --- | --- | --- | --- | --- |
| RPR: $L_1 L^{-1}$ | $L$ | $L_1$ | $L_2$ | $H$ | $H$ |
| 0.10 | 10 m | 1.0 m | 9.0 m | 3.8 m | 6.0 m |
| 0.15 | 10 m | 1.5 m | 8.5 m | 3.8 m | 6.0 m |
| 0.20 | 10 m | 2.0 m | 8.0 m | 3.8 m | 6.0 m |
| 0.22 | 10 m | 2.2 m | 7.8 m | 3.8 m | 6.0 m |
| 0.25 | 10 m | 2.5 m | 7.5 m | 3.8 m | 6.0 m |
| 0.30 | 10 m | 3.0 m | 7.0 m | 3.8 m | 6.0 m |
| 0.35 | 10 m | 3.5 m | 6.5 m | 3.8 m | 6.0 m |
| 0.40 | 10 m | 4.0 m | 6.0 m | 3.8 m | 6.0 m |
| 0.45 | 10 m | 4.5 m | 5.5 m | 3.8 m | 6.0 m |

500 000 and during four steps of mesh refinement, it finally reached 1 500 000. The average air temperature (0.00) inside the greenhouse relating to the third level of mesh refinement (about 1 000 000 cells) showed a satisfactory precision compared with the fourth level (1 250 000 cells). Increasing the cells number to 1 500 000 (fifth grid) did not show a considerable improvement in the accuracy of the results. Therefore, considering the calculation burden and the guarantee of simulation accuracy, the same meshing method was also adopted for other contrastive cases in the parameter investigation, and the total grid number was approximately 1 000 000.

# 3. Results

## 3.1. The model validation

To validate the accuracy of CFD numerical model, the simulated temperature was compared with the experimental temperature under the same environmental conditions. The average air temperature of each measuring point (P1–P6) was taken as the measurement result and compared with the simulated average air temperature [45]. Figure 4 shows numerical and experimental data on the variation of the internal averaged air temperature. The numerical results of the whole simulation period were compared with the experimental results. The average absolute temperature difference measured and simulated was 0.22°C. The average relative error between the measured results and the numerical results was about 9.5%. This indicated that the proposed mathematical model describes the processes occurring in the greenhouse with a high level of accuracy. The numerical results were lower than the measured values in the daytime due to the thermocouple shell being heated by solar radiation and having a certain heat preservation function. The numerical results were slightly higher than the measured values after the heat preservation quilt was expanded. This is because the heat preservation quilt is colder than the numerical model [25].

## 3.2. Thermal environment analysis

During the cultivation of crops in winter, the temperature distribution of the internal air, soil and north wall should be monitored carefully. The air environment directly affects the growth of crops, while the soil environment directly affects the growth of crop roots. The north wall, which is the most important heat storage-release enclosure of CSG, directly affects the temperature stability of the internal air. Figure 5 shows the temperature distribution at the intermediate cross-section of the greenhouse of different RPR at several typical moments. The labelled temperature in the figure is the average body air temperature in the greenhouse. It can be seen from figure 5a,b that the occurrence time of the maximum daytime temperature in the greenhouse will be delayed (from 14.00 to 15.00) when RPR increases to 0.3. In addition, RPR also has a great effect on the spatial distribution of daytime temperature inside the greenhouse. It can be seen from figure 5c that RPR has relatively

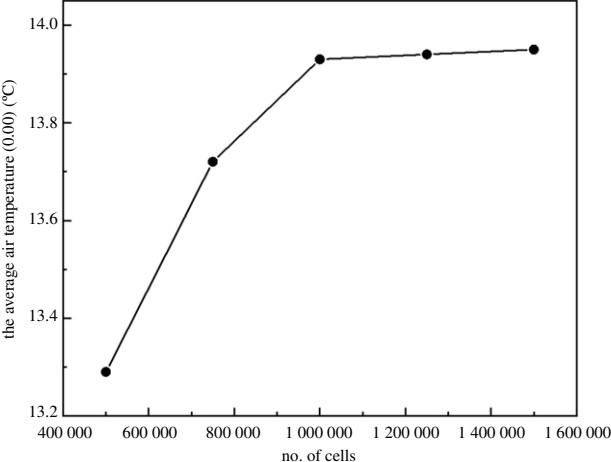

**Figure 3.** Computed average air temperature (0.00) inside the greenhouse based on the different grid of the domain.

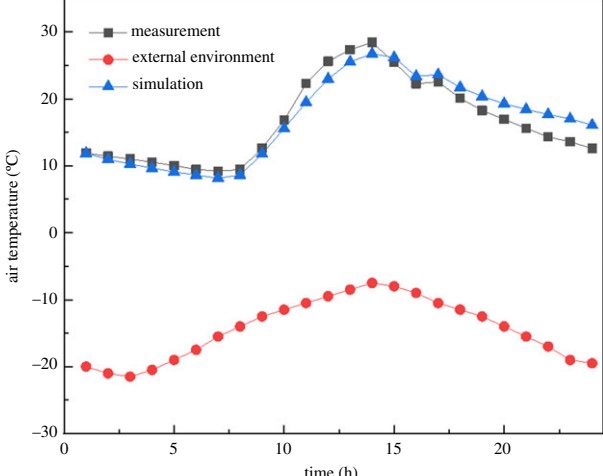

**Figure 4.** Internal air temperatures obtained via experiments and simulations.

**Table 2.** The thermophysical properties of the envelope materials in the solar greenhouse.

| material | density (kg m$^{-3}$) | specific heat (J kg$^{-1}$ K$^{-1}$) | Thermal conductivity (W m$^{-1}$ K$^{-1}$) |
|---|---|---|---|
| fly ash brick | 1600 | 1051.1 | 0.5 |
| polystyrene board | 30 | 2414.8 | 0.041 |
| board | 550 | 2510 | 0.29 |
| grass mat | 300 | 1680 | 0.13 |
| soil | 1700 | 1010 | 0.85 |
| PO plastic film | 950 | 1600 | 0.19 |
| heat preservation quilt | 70 | 840 | 0.05 |

little effect on the minimum temperature at night in each part of the greenhouse, which will be analysed in detail in the next section.

Figure 6 shows the hourly variation of the average air temperature inside the greenhouse with different RPR. The temperature distributions are significantly different during a day under different RPR. The effect of RPR on the thermal environment of the solar greenhouse was analysed from the following aspects: the maximum and the minimum temperatures of the air, soil surface and north

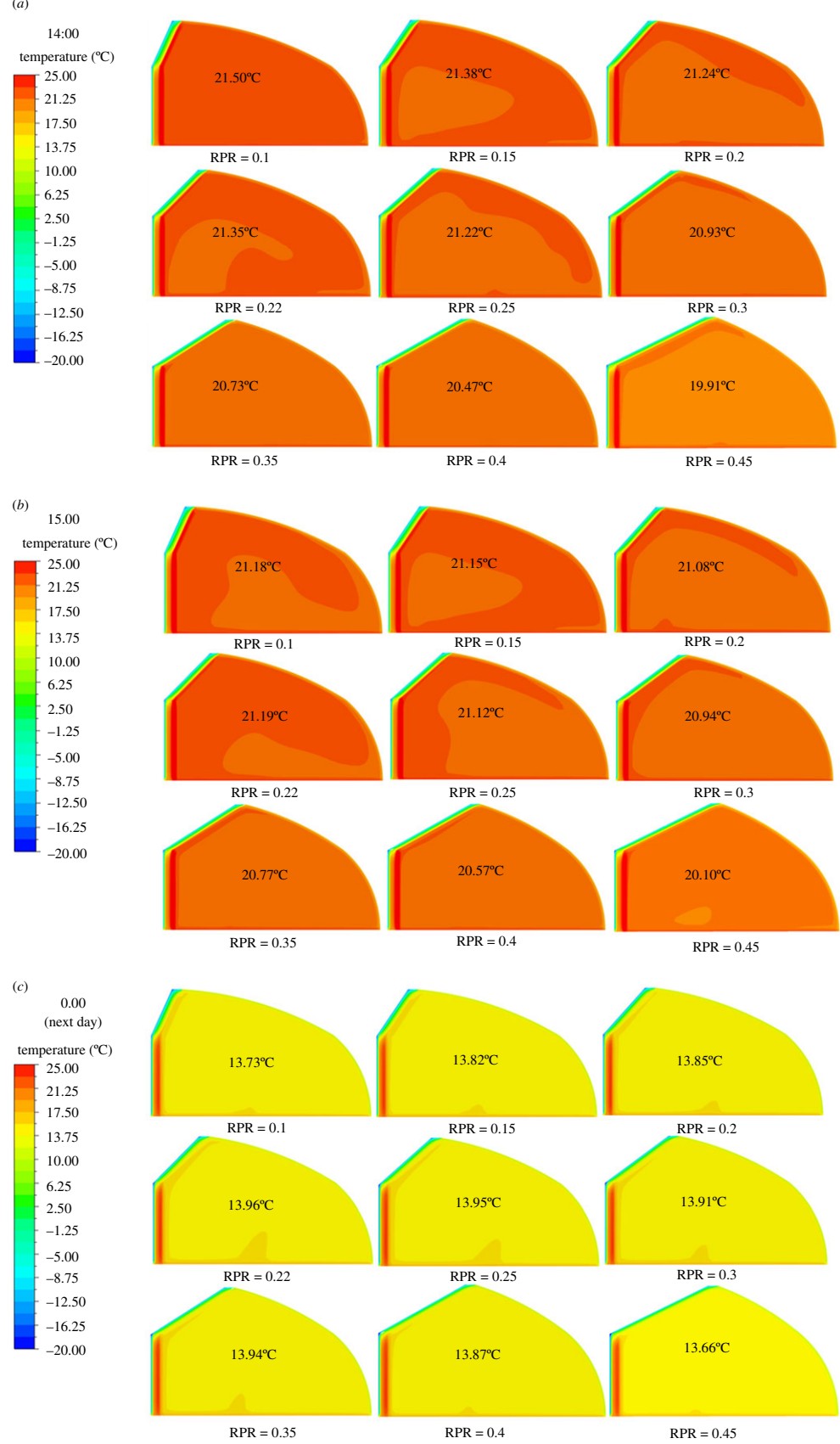

**Figure 5.** Temperature distribution at the intermediate cross-section of the greenhouse of different RPR at several typical moments: (*a*) 14.00, (*b*) 15.00 and (*c*) 0.00 (next day).

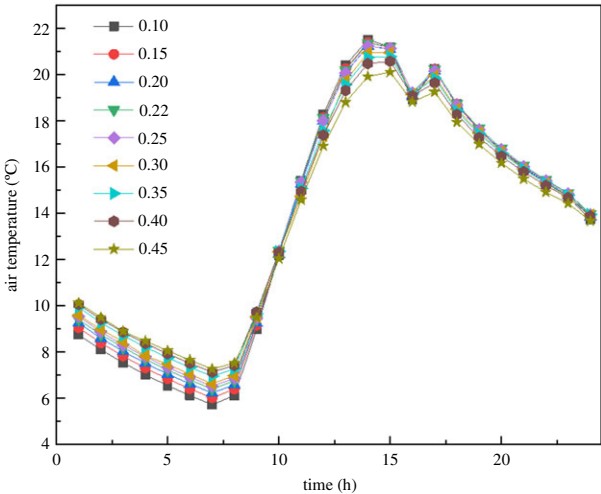

**Figure 6.** Hourly variation of the internal temperature averaged in the air space inside the greenhouse with different RPR.

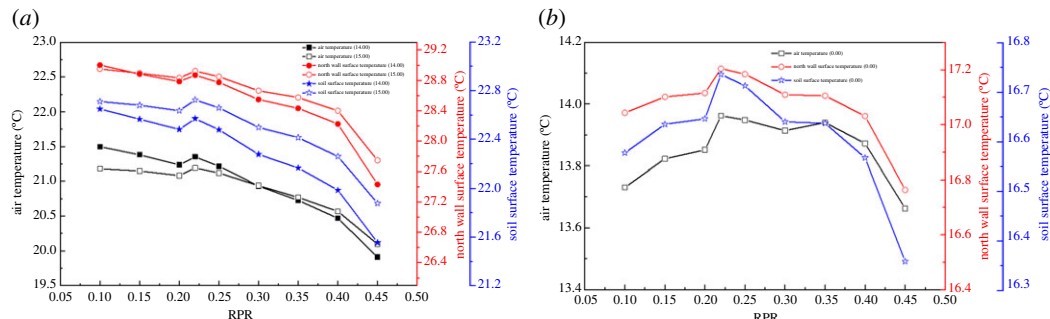

**Figure 7.** Effect of RPR on the maximum and minimum temperatures of the air and solid surfaces: (*a*) maximum temperature and (*b*) minimum temperature.

wall surface; the temperature distribution of the soil with depth; and the temperature distribution of the north wall with depth.

### 3.2.1. The maximum and minimum temperatures of air and solid surfaces

Figure 7 shows the effect of RPR on the maximum and minimum temperatures of the air and solid surfaces. Figure 7*a* shows that the maximum air temperature and the occurrence time varied as RPR increased, and the maximum temperature had an overall decreasing trend. The amount of solar energy that entered the greenhouse was reduced because the south roof area decreased as RPR increased. The maximum air temperature slightly increased when RPR changed to 0.22, which indicates that the amount of solar energy that entered the greenhouse and the heat released to the outside reached equilibrium. It is worth noting that the emergence time of the maximum air temperature alternated when RPR was around 0.3. In other words, the maximum air temperature occurred at 14.00 if RPR was less than 0.3, because the south roof area was relatively larger, allowing more solar energy to enter the greenhouse in the forenoon. The air temperature rise was larger and the rate of temperature rise was quicker during the daytime. An increase in the amount of solar energy was not able to offset the cooling effect, resulting in a maximum internal air temperature appearing at 14.00. However, the amount of solar energy that entered the greenhouse was relatively lower when RPR was greater than 0.3. The air temperature rise was lower and the rate of temperature rise was slower. Correspondingly, the area of heat dissipation also decreased, which was able to offset the cooling effect. The heat stored in the greenhouse was difficult to spread out, resulting in a maximum internal air temperature at 15.00. Hence, overall, the maximum air temperature declined as RPR increased, but the maximum air temperature increased slightly when the RPR increased to a certain value, and the occurrence time of the maximum air temperature will be delayed when the RPR increases to a saturation value. Figure 7*a* also shows the effect of RPR on the maximum

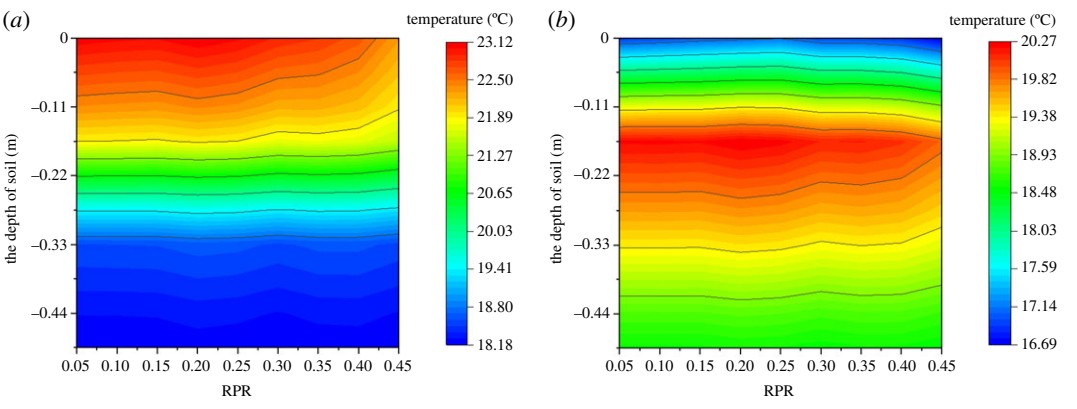

**Figure 8.** Effect of RPR on the temperature at different depths of the soil at typical moments: (*a*) 15.00 and (*b*) 0.00 (next day).

temperature of the soil surface. The occurrence time of the maximum temperature of the soil surface was delayed relative to the air when the RPR was less than 0.3, because the soil was the largest heat storage body in the greenhouse, which had stable thermodynamic properties and large specific heat capacity. It can be seen from figure 7*a* that the maximum temperature on the north wall surface alternated when the RPR was around 0.15. The maximum temperature of the north wall surface appeared at 15.00 when the RPR was greater than 0.15. Compared with the soil, the north wall had better thermal storage and release properties, which directly affected the temperature variation and stability of the internal air. Therefore, the maximum temperature of the north wall also captured this alternative phenomenon. As RPR increased, the south roof area decreased and the north roof area increased, which suggests that the amount of solar energy that entered the greenhouse decreased and the heat dissipation area decreased. On the contrary, the heat preservation area increased and the heat preservation effect was strengthened. It does not easily dissipate the heat stored in the greenhouse, which is why maximum air temperature alternates occurred. It is worth noting that figure 7*a* can be divided into three RPR intervals according to the slope of the curve. They were 0.1–0.22, 0.22–0.4 and 0.4–0.45, respectively. This suggests that the decline rate of the maximum temperature of each part in different RPR intervals increased as RPR increased. Because as RPR increases, the amount of solar energy which entered the greenhouse decreases, the area of heat dissipation decreases, the convection heat transfer between the enclosure and the internal air was enhanced.

The temperature at 0.00 (next day) can represent the minimum air temperature in the greenhouse because there is no energy input into the greenhouse for a long time at night. The variation trend of night temperature was mainly determined by the heat storage during the day and the insulation performance of the enclosure structure. Figure 7*b* shows the variation of the minimum temperature as RPR increased. The maximum air temperature difference was 0.3°C. The minimum temperature of each part of the greenhouse peaked when RPR changed to 0.22, due to the retention effect of air temperature during the day. The maximum temperature difference of soil surface and north wall surface was 0.38°C and 0.44°C, respectively, as RPR increased at night. Compared with the variation of the maximum temperature, the difference in the minimum temperature of each part was not obvious as RPR increased. The minimum air temperature was affected primarily by natural convection between the soil, the north wall and the internal air. The soil and the north wall were the main heat-release bodies, which directly affected the variation trend of the minimum air temperature. RPR had minimal effect on the minimum temperature of each part of the greenhouse at night because the size of the soil and north wall almost had no change with the variation of RPR. The internal air temperature at night was mainly maintained by the greenhouse envelope.

### 3.2.2. Solid interior temperature

Figure 8 shows the effect of RPR on the temperature at different depths of the soil. It can be seen from figure 8*a* that the soil temperature decreases as the soil depth increases. The temperature distribution was greatly affected by RPR when the soil depth was less than 0.17 m. However, when the soil depth was greater than 0.17 m, the soil temperature was unchanged despite the variation of RPR. Figure 8*b* shows the effect of RPR on the temperature at different depths of the greenhouse soil at 0.00 (next day). It can be seen that the soil temperature first increased then decreased as the soil depth increased,

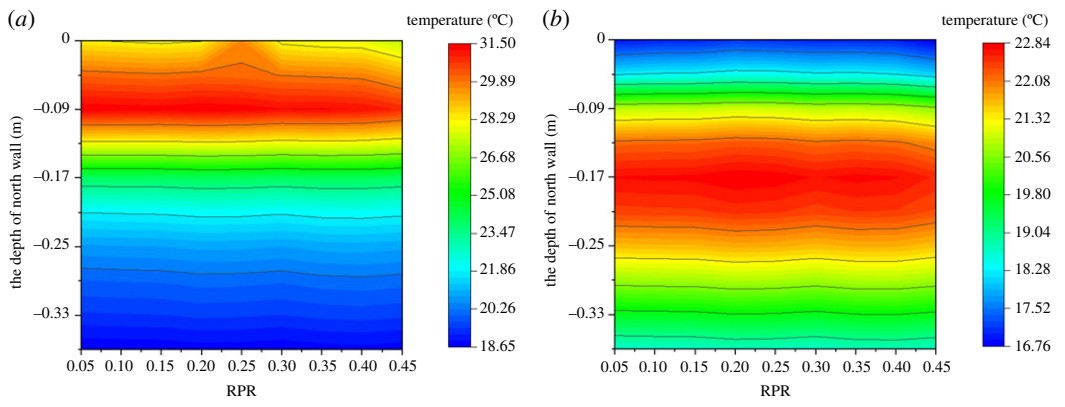

**Figure 9.** Effect of RPR on the temperature at different depths of the north wall at typical moments: (*a*) 15.00 and (*b*) 0.00 (next day).

and the maximum temperature occurred at about 0.17 m below the soil surface. The heat storage layer of the soil became thinner when RPR was greater than 0.3. Furthermore, the average soil temperature decreased as RPR increased during the whole day. This indicated that the amount of solar energy absorbed by the soil decreased during the day. The heat storage layer of the soil became thinner, while the convective heat transfer between the soil and the internal air became stronger, this maintaining the stability of air temperature at night. The results showed that the average temperature of the soil in the greenhouse decreased gradually during the day and the heat storage layer of the soil became thinner at night as RPR increased.

Figure 9 shows the effect of RPR on the temperature at different depths of the north wall. It can be seen from figure 9a that the temperature first increased then decreased as the north wall depth increases. The maximum temperature was about 0.1 m away from the north wall surface, and the difference of the maximum temperatures between the RPR of 0.22 and 0.45 was 0.57°C. When the depth of the north wall was greater than 0.1 m, the temperature of the north wall remained unchanged despite the variation of RPR. Figure 9b shows the effect of RPR on the temperature at different depths of the north wall at 0.00 (next day). It can be seen that the temperature first increased then decreased as the north wall depth increased. The maximum temperature was about 0.17 m away from the north wall surface, and the difference of the maximum temperatures between the RPR of 0.22 and 0.45 was 0.27°C. The heat storage layer of the north wall remained unchanged despite the variation of RPR. This indicated that the RPR has minimal effect on the temperature distribution of the north wall. Compared with the soil, the north wall was more capable of regulating natural convection with the internal air. The north wall had a strong heat exchange ability and good thermal storage-release property. The results showed that the average temperature of the north wall in the greenhouse decreased gradually during the day and the heat storage layer of the north wall remained constant at night despite the increase of RPR.

## 3.3. The primary cause of the thermal environment: energy analysis

### 3.3.1. Solar energy input

Figure 10 shows the effect of the south roof area, north roof area and solar energy interception with different RPR. It is worth noting that the variations of the three influence factors almost present linear trends as RPR increases. Figure 11 shows the effect of absorbed solar energies of the north wall and the soil with different RPR. By contrast, the solar energy absorbed by the north roof is negligible compared with those of the soil and the north wall. The amount of solar energy absorbed by the soil and the north wall decreased gradually as RPR increased. Moreover, the soil absorbed more solar energy than the north wall. The results also suggested that the variation of RPR influenced the absorption of solar energy by soil (decrease by 5.90%) compared with the north wall (decrease by 5.15%). This was because the soil area inside the greenhouse was large and the thermal property was stable.

The relationships between the RPR $x$ and solar energy that entered the greenhouse $S_{ei}$ were described by the following correlation equation (3.1) when the RPR is 0.1–0.45. And the relationships between the

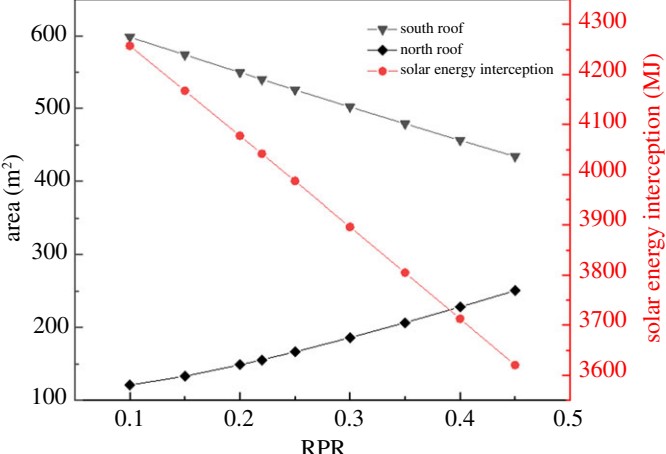

**Figure 10.** Effect of the south roof area, north roof area, and solar energy interception with different RPR.

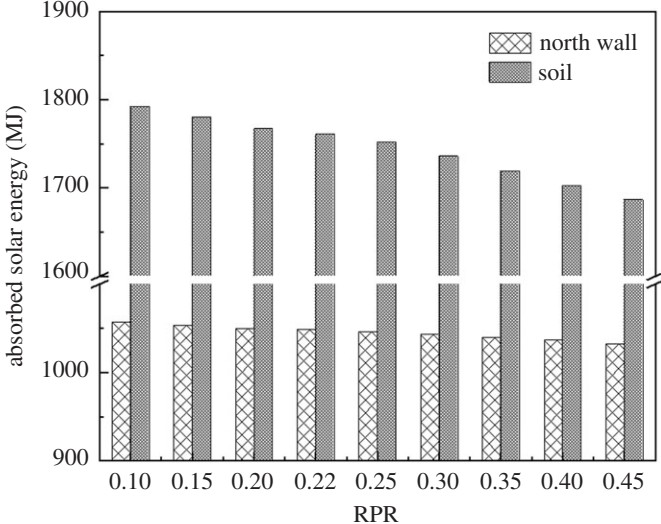

**Figure 11.** Effect of absorbed solar energy of the north wall and soil with different RPR.

RPR $x$ and solar energy that absorbed by the north wall $S_{a-nw}$, solar energy that absorbed by soil $S_{a-s}$ were described by the following correlation equations (3.2) and (3.3).

$$S_{ei} = -1819.29x + 4440.88, \ R^2 = 0.99, \tag{3.1}$$

$$S_{a-nw} = -69.28x + 1063.84, \ R^2 = 0.99 \tag{3.2}$$

and

$$S_{a-s} = -307.57x + 1826.96, \ R^2 = 0.99. \tag{3.3}$$

### 3.3.2. Heat-released capacity

Figure 12 shows the effect of RPR on released heat energy at different moments. It can be seen from figure 12*a,b* that the heat was mainly lost through the south roof. As RPR increased, the heat lost through the south roof decreased by 181.99 MJ (30.12%) and 217.75 MJ (28.90%) during the day and at night, respectively. The stored heats in the north wall and the soil play an important role in the heat-release process. The north wall releases the most heat during the day, while the soil releases the most heat at night. Since the area of the soil and north wall remains the same, the heat released from them fluctuates only slightly. It is worth noting that the heat released from the north roof varies over time. When the RPR was less than 0.25 (during the day) and 0.22 (at night), the north roof releases heat into the greenhouse, and the released heat decreases as RPR increases. When the RPR was greater than 0.25 (during the day) and 0.22 (at night), the greenhouse releases heat to the outside through the

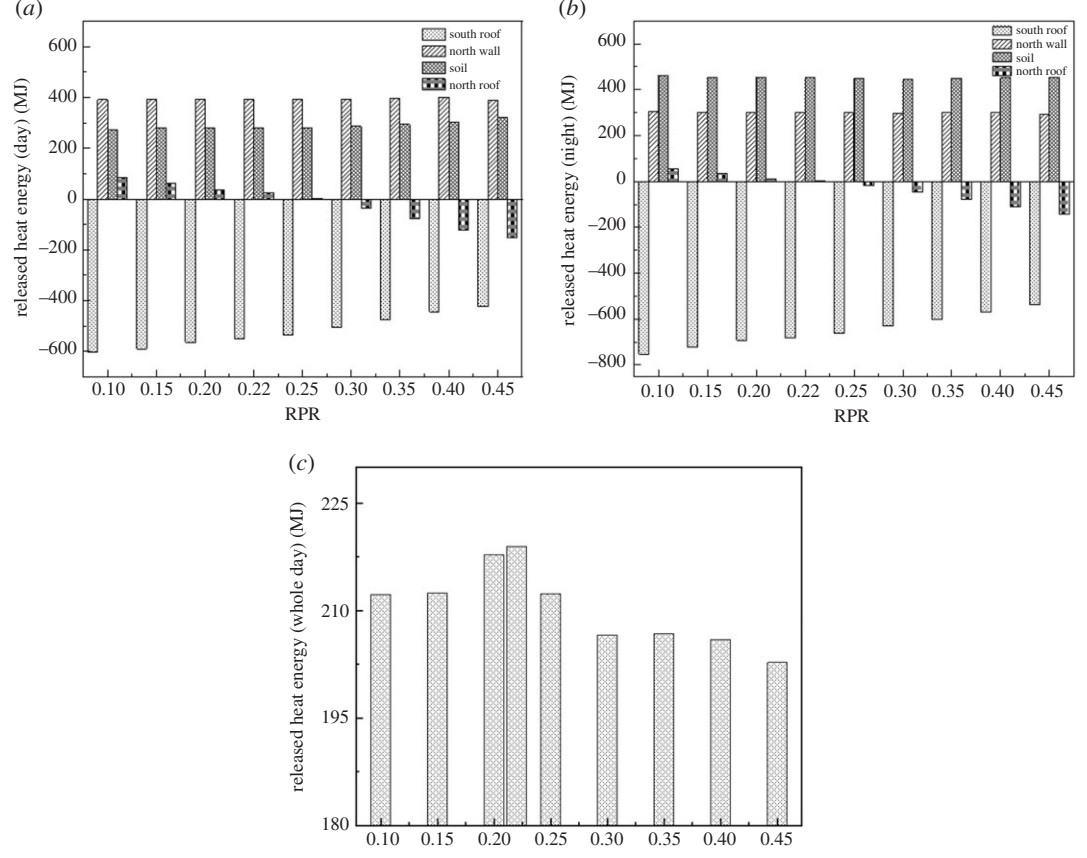

**Figure 12.** Effect of RPR on released heat energy at different moments: (*a*) day, (*b*) night and (*c*) whole day.

north roof, and the released heat increases as RPR increases. Figure 12*c* shows the influence of the enclosure structure on the heat storage-release during the whole day. The variation trend is consistent with the variation trend of the minimum internal air temperature. The results showed that the heat released to the inside greenhouse by the north roof is less than the heat released to the outside through the north roof when the RPR increases to a certain value. Therefore, the area of the north roof should not be excessive.

The relationships between the released heat energy $E_{rh}$ and RPR $x$ were described by the following correlation equation (3.4), and the specific numerical details are shown in the table 3.

$$E_{rh} = a + B_1 x + B_2 x^2 + B_3 x^3 + B_4 x^4. \tag{3.4}$$

# 4. Conclusion

A mathematical model was established by using the method of combining CFD numerical simulation with experiments in this paper, then used to analyse the effect of the RPR on the thermal environment of the solar greenhouse. The maximum air temperature had an overall decreasing trend during the daytime as RPR increases. However, the maximum air temperature increased slightly when RPR increased to 0.22. The occurrence time of the maximum air temperature inside the greenhouse will be delayed when RPR increases to 0.3. The maximum temperature of the north wall also captured this alternative phenomenon. The decline rate of the maximum temperature of each part in different RPR intervals increased as RPR increased. RPR has a relatively small effect on the minimum temperature of each greenhouse part during the night.

Mathematical models of the relationships between RPR and solar energy that entered the greenhouse, solar energy that was absorbed by the north wall, solar energy that was absorbed by the soil, and the released heat energy of the enclosure structures of the greenhouse were established. The amount of solar energy that entered the greenhouse was reduced as RPR increased. The variation of RPR has a

**Table 3.** The specific numerical details of the relationships between the released heat energy $E_{rh}$ and RPR $x$.

| greenhouse envelope | time | $a$ | $B_1$ | $B_2$ | $B_3$ | $B_4$ | $R^2$ |
|---|---|---|---|---|---|---|---|
| south roof | day | $-668.3945 \pm 4.4005$ | $548.5913 \pm 15.1591$ | 0 | 0 | 0 | 0.99 |
| | night | $-816.3357 \pm 1.1422$ | $618.9108 \pm 3.9348$ | 0 | 0 | 0 | 0.99 |
| north roof | day | $1713481 \pm 8.3477$ | $-712.9040 \pm 28.7568$ | 0 | 0 | 0 | 0.99 |
| | night | $123.7489 \pm 4.8884$ | $-581.2796 \pm 16.8399$ | 0 | 0 | 0 | 0.99 |
| soil | day | $283.1134 \pm 6.1848$ | $-115.7389 \pm 48.8621$ | $424.4669 \pm 86.6294$ | 0 | 0 | 0.96 |
| | night | $474.3387 \pm 2.4947$ | $-161.1863 \pm 19.7090$ | $262.7152 \pm 34.9428$ | 0 | 0 | 0.90 |
| north wall | day | $339.79163 \pm 15.1926$ | $1030.4782 \pm 280.9753$ | $-7022.9709 \pm 1764.1245$ | $19798.2525 \pm 4541.0301$ | $-19383.3818 \pm 4107.2962$ | 0.94 |
| | night | $292.1873 \pm 12.2834$ | $314.4969 \pm 229.0212$ | $-2583.0913 \pm 1437.9244$ | $7709.8739 \pm 3701.3588$ | $-7806.0848 \pm 3347.8256$ | 0.92 |

greater influence on the capacity of the soil (decrease by 5.90%) than the north wall (decrease by 5.15%) to absorb solar energy. As RPR increases, the heat storage layer of the soil gradually becomes thinner, but the heat storage layer of the north wall basically remains unchanged. The effect of reducing heat loss from the south roof is greater as RPR increased. Notably, when RPR was less than 0.25 (during the daytime) and 0.22 (during the night), the north roof releases heat into the greenhouse, and the released heat decreases as RPR increases. When RPR was larger than 0.25 (during the daytime) and 0.22 (during the night), the heat is released to the outside through the north roof, and the released heat increases as RPR increases. That is to say, the direction in which the heat is released through the north roof changes when RPR increases to a certain value. The future work will focus on using the model to further optimize the structural design parameters of the Chinese solar greenhouse, such as other important parameters, the ratio of ridge height to span length. In addition, a humidity sub-model is added to this model to further discuss the related problems of high relative humidity when the greenhouse is closed during the night.

Ethics. This study does not involve the ethical approval and informed consent.

Data accessibility. A dimensionless parameter coupling the south and north roofs on the thermal environment of Chinese solar greenhouse: Ridge position ratio, Dryad, Dataset, https://doi.org/10.5061/dryad.44j0zpccb [46].

Authors' contributions. X.W., X.L., H.X. and Y.L. were involved in conceptualization. X.W., X.L., Y.L. and X.Y. were involved in data curation. X.W., X.L., Y.L. and X.Y. were involved in formal analysis. Y.L. was involved in funding acquisition. X.W., Y.L., X.Y. and H.X. were involved in investigation. Y.L., X.Y. and H.X. were involved in methodology. H.X. was involved in project administration. X.Y., H.X. and T.L. were involved in resources. X.W., X.L. and Y.L. were involved in software. X.W. was involved in supervision. X.W., Y.L. and X.Y. were involved in validation. X.W. was involved in writing the original draft. Y.L. was involved in writing the review and editing. X.L. had a role in the study design, data collection and analysis, decision to publish or preparation of the manuscript.

Competing interests. The authors declare that they have no known competing financial interests or personal relationships that could have appeared to influence the work reported in this paper.

Funding. This work was supported by The National Key Research and Development Program of China (2019YFD1001902) under the auspices of The Ministry of Science and Technology of the People's Republic of China.

Acknowledgements. This work was supported by The National Key Research and Development Program of China (2019YFD1001902) under the auspices of The Ministry of Science and Technology of the People's Republic of China. We thank all members of National & Local Joint Engineering Research Center of Northern Horticultural Facilities Design & Application Technology (Liaoning) for their help.

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
