## [Peer Review File · Royal Society Open Science]

Review History

RSOS-201707.R0 (Original submission)

Review form: Reviewer 1

Is the manuscript scientifically sound in its present form?

No

Are the interpretations and conclusions justified by the results?

Yes

Is the language acceptable?

Yes

Do you have any ethical concerns with this paper?

No

Have you any concerns about statistical analyses in this paper?

No

Recommendation?

Major revision is needed (please make suggestions in comments)

Comments to the Author(s)

The present paper investigates the performance of a Chinese Solar Greenhouse (CSG) regarding the efficient exploitation of solar energy. The results are important since CSG is a very common, economically significant greenhouse type. The paper examines the effect of the length of the north roof on the indoor temperature of a closed greenhouse without crop. Other possibly important parameters, such as the ratio of ridge-height to span-length, have not been investigated. Problems related to high relative humidity when the greenhouse is closed during the night have not been discussed. Such points, if not included in a revised version of the current paper, should be at least mentioned as future research plans in the Conclusions. Moreover, I suggest that the fact that the results concern an empty closed greenhouse should be explained in the abstract.

A few additional specific comments follow:

- 1) The title is too long and possibly confusing. I suggest a simpler one: e.g., Effect of the ridge position on the thermal environment of the Chinese Solar Greenhouse.
 - 2) Symbols are not well represented in the used font, so they must be corrected.
 - 3) Ridge Position Ratio (RPR) is defined in the Introduction. The definition should be transferred to the Materials and Methods section.
 - 4) The Materials and Methods section is short, so sub-sections are not necessary.
 - 5) The $k-\epsilon$ model and the Boussinesq approximation are well-known models so no detailed explanation is required. A relevant reference could be sufficient.
 - 6) Mesh characteristics could be critical for the quality of the numerical results. For this reason, detailed information regarding the mesh should be provided in the Materials and Methods section.
 - 7) The boundary conditions are not described besides a short statement that "The standard wall function is applied to near-wall processing". A paragraph is necessary to address in detail the boundary conditions on all areas closing the greenhouse, i.e., north and south roof, soil, north wall, etc.
 - 8) Figure 4: Replace the word interface with cross-section.
 - 9) Figure 9: Do the fitting curves correspond to equations 10-12?
- The language needs improvements.

Review form: Reviewer 2

Is the manuscript scientifically sound in its present form?

No

Are the interpretations and conclusions justified by the results?

Yes

Is the language acceptable?

Yes

Do you have any ethical concerns with this paper?

No

Have you any concerns about statistical analyses in this paper?

No

Recommendation?

Major revision is needed (please make suggestions in comments)

Comments to the Author(s)

The manuscript entitled "A dimensionless parameter coupling the south and north roofs on the thermal environment of Chinese solar greenhouse: Ridge position ratio" is very interesting and might be suitable for the journal of Royal Society Open Science. Based on this article's review, the authors need to address the followings issues before considering them for publication.

Major concerns

Literature review: The authors applied a different approach to analyzing the thermal performance of CSG based on the variation of the north roof dimension. However, I found a significant lacking in terms of literature review; several researchers used a similar CFD based model to analyze the thermal performance of CSG. Authors need to state the novelty of their approach to studying thermal performance. The authors also need to state the advantage of their approach compared to the model based on the energy balance equation.

For example: Ahamed, M. S., Guo, H., & Tanino, K. (2018). Development of a thermal model for simulation of supplemental heating requirements in Chinese-style solar greenhouses. *Computers and Electronics in Agriculture*, 150, 235-244.

Unit for Temperature: I see the authors used K as a unit for Temperature, which is not usual for journal publication. I would recommend presenting the results in C.

PRR: I would recommend drawing the line in figure 2 to show the visualization for PRP.

Graph: The caption for the graph looks very confusing. Please revise the caption; if the authors like to combine several figures, please present it as a single picture. Figure 4: I don't see any significant difference in temperature variation for different RPR. Authors need to explain the reason for this; if this is the case, then the significance of RPR is not very crucial for the thermal performance of the CSG.

Decision letter (RSOS-201707.R0)

Dear Professor Li

The Editors assigned to your paper RSOS-201707 "A dimensionless parameter coupling the south and north roofs on the thermal environment of Chinese solar greenhouse: Ridge position ratio" have now received comments from reviewers and would like you to revise the paper in accordance with the reviewer comments and any comments from the Editors. Please note this decision does not guarantee eventual acceptance.

Please submit your revised manuscript and required files (see below) no later than 21 days from today's (ie 18-Feb-2021) date. Note: the ScholarOne system will 'lock' if submission of the revision is attempted 21 or more days after the deadline. If you do not think you will be able to meet this deadline please contact the editorial office immediately.

on behalf of Professor R. Kerry Rowe (Subject Editor)
openscience@royalsociety.org

Associate Editor Comments to Author:

Thank you for your patience while we sought reviewers for your work - unfortunately, and no doubt owing to the ongoing COVID crisis, the journal has struggled to find referees. Happily, however, two have been kind enough to return reports. Each raises substantial, but not irredeemable, concerns regarding your study - there are a number of scientific concerns as well as in relation to the presentation (clarity of figures) and the quality of the written English. We would strongly recommend you take advantage of a professional language editing service before you resubmit any revision - some examples, including those the journal has a discount with, are at <https://royalsociety.org/journals/authors/benefits/language-editing/>. Any revision must include not only a tracked changes version but also a clear point-by-point response document detailing the changes made. We will invite the reviewers to assess the revision in due course.

Reviewer comments to Author:
Reviewer: 1
Comments to the Author(s)

The present paper investigates the performance of a Chinese Solar Greenhouse (CSG) regarding the efficient exploitation of solar energy. The results are important since CSG is a very common, economically significant greenhouse type. The paper examines the effect of the length of the north roof on the indoor temperature of a closed greenhouse without crop. Other possibly important parameters, such as the ratio of ridge-height to span-length, have not been investigated. Problems related to high relative humidity when the greenhouse is closed during the night have not been discussed. Such points, if not included in a revised version of the current

paper, should be at least mentioned as future research plans in the Conclusions. Moreover, I suggest that the fact that the results concern an empty closed greenhouse should be explained in the abstract.

A few additional specific comments follow:

- 1) The title is too long and possibly confusing. I suggest a simpler one: e.g., Effect of the ridge position on the thermal environment of the Chinese Solar Greenhouse.
- 2) Symbols are not well represented in the used font, so they must be corrected.
- 3) Ridge Position Ratio (RPR) is defined in the Introduction. The definition should be transferred to the Materials and Methods section.
- 4) The Materials and Methods section is short, so sub-sections are not necessary.
- 5) The k - ϵ model and the Boussinesq approximation are well-known models so no detailed explanation is required. A relevant reference could be sufficient.
- 6) Mesh characteristics could be critical for the quality of the numerical results. For this reason, detailed information regarding the mesh should be provided in the Materials and Methods section.
- 7) The boundary conditions are not described besides a short statement that "The standard wall function is applied to near-wall processing". A paragraph is necessary to address in detail the boundary conditions on all areas closing the greenhouse, i.e., north and south roof, soil, north wall, etc.
- 8) Figure 4: Replace the word interface with cross-section.
- 9) Figure 9: Do the fitting curves correspond to equations 10-12?
The language needs improvements.

Reviewer: 2

Comments to the Author(s)

The manuscript entitled "A dimensionless parameter coupling the south and north roofs on the thermal environment of Chinese solar greenhouse: Ridge position ratio" is very interesting and might be suitable for the journal of Royal Society Open Science. Based on this article's review, the authors need to address the followings issues before considering them for publication.

Major concerns

Literature review: The authors applied a different approach to analyzing the thermal performance of CSG based on the variation of the north roof dimension. However, I found a significant lacking in terms of literature review; several researchers used a similar CFD based model to analyze the thermal performance of CSG. Authors need to state the novelty of their approach to studying thermal performance. The authors also need to state the advantage of their approach compared to the model based on the energy balance equation.

For example: Ahamed, M. S., Guo, H., & Tanino, K. (2018). Development of a thermal model for simulation of supplemental heating requirements in Chinese-style solar greenhouses. *Computers and Electronics in Agriculture*, 150, 235-244.

Unit for Temperature: I see the authors used K as a unit for Temperature, which is not usual for journal publication. I would recommend presenting the results in C.

PRR: I would recommend drawing the line in figure 2 to show the visualization for PRP.

Graph: The caption for the graph looks very confusing. Please revise the caption; if the authors like to combine several figures, please present it as a single picture. Figure 4: I don't see any significant difference in temperature variation for different RPR. Authors need to explain the reason for this; if this is the case, then the significance of RPR is not very crucial for the thermal performance of the CSG.

===PREPARING YOUR MANUSCRIPT===

===PREPARING YOUR REVISION IN SCHOLARONE===

<https://royalsociety.org/journals/authors/author-guidelines/#supplementary-material> to include a suitable title and informative caption. An example of appropriate titling and captioning may be found at https://figshare.com/articles/Table_S2_from_Is_there_a_trade-off_between_peak_performance_and_performance_breadth_across_temperatures_for_aerobic_sc_ope_in_teleost_fishes_/3843624.

Author's Response to Decision Letter for (RSOS-201707.R0)

See Appendix A.

RSOS-201707.R1 (Revision)

Review form: Reviewer 1

Is the manuscript scientifically sound in its present form?

Yes

Are the interpretations and conclusions justified by the results?

Yes

Is the language acceptable?

Yes

Do you have any ethical concerns with this paper?

No

Have you any concerns about statistical analyses in this paper?

No

Recommendation?

Accept as is

Comments to the Author(s)

I have no further comments on this paper.

Review form: Reviewer 2

Is the manuscript scientifically sound in its present form?

Yes

Are the interpretations and conclusions justified by the results?

Yes

Is the language acceptable?

Yes

Do you have any ethical concerns with this paper?

No

Have you any concerns about statistical analyses in this paper?

No

Recommendation?

Accept with minor revision (please list in comments)

Comments to the Author(s)

The article could be accepted with minor revision.

Please state the advantages of your model over CSGHEAT and CFD in the introduction section.

Decision letter (RSOS-201707.R1)

Dear Professor Li

On behalf of the Editors, we are pleased to inform you that your Manuscript RSOS-201707.R1 "Effect of the ridge position ratio on the thermal environment of the Chinese Solar Greenhouse" has been accepted for publication in Royal Society Open Science subject to minor revision in accordance with the referees' reports. Please find the referees' comments along with any feedback from the Editors below my signature.

Please submit your revised manuscript and required files (see below) no later than 7 days from today's (ie 12-Apr-2021) date. Note: the ScholarOne system will 'lock' if submission of the revision is attempted 7 or more days after the deadline. If you do not think you will be able to meet this deadline please contact the editorial office immediately.

on behalf of R. Kerry Rowe (Subject Editor)
openscience@royalsociety.org

Associate Editor Comments to Author:
Comments to the Author:
Please address the final comments raised by the referee before resubmitting.

Reviewer comments to Author:
Reviewer: 1

Comments to the Author(s)
I have no further comments on this paper.

Reviewer: 2

Comments to the Author(s)
The article could be accepted with minor revision.
Please state the advantages of your model over CSGHEAT and CFD in the introduction section.

===PREPARING YOUR MANUSCRIPT===

===PREPARING YOUR REVISION IN SCHOLARONE===

- An editable file of each table (.doc, .docx, .xls, .xlsx, or .csv).
- An editable file of all figure and table captions.

- Any electronic supplementary material (ESM).
- If you are requesting a discretionary waiver for the article processing charge, the waiver form must be included at this step.
- If you are providing image files for potential cover images, please upload these at this step, and inform the editorial office you have done so. You must hold the copyright to any image provided.
- A copy of your point-by-point response to referees and Editors. This will expedite the preparation of your proof.

- Ensure that your data access statement meets the requirements at <https://royalsociety.org/journals/authors/author-guidelines/#data>. You should ensure that you cite the dataset in your reference list. If you have deposited data etc in the Dryad repository, please only include the 'For publication' link at this stage. You should remove the 'For review' link.
- If you are requesting an article processing charge waiver, you must select the relevant waiver option (if requesting a discretionary waiver, the form should have been uploaded at Step 3 'File upload' above).
- If you have uploaded ESM files, please ensure you follow the guidance at <https://royalsociety.org/journals/authors/author-guidelines/#supplementary-material> to include a suitable title and informative caption. An example of appropriate titling and captioning may be found at https://figshare.com/articles/Table_S2_from_Is_there_a_trade-off_between_peak_performance_and_performance_breadth_across_temperatures_for_aerobic_scope_in_teleost_fishes_/3843624.

Author's Response to Decision Letter for (RSOS-201707.R1)

See Appendix B.

Decision letter (RSOS-201707.R2)

Dear Professor Li,

I am pleased to inform you that your manuscript entitled "Effect of the ridge position ratio on the thermal environment of the Chinese Solar Greenhouse" is now accepted for publication in Royal Society Open Science.

on behalf of R. Kerry Rowe (Subject Editor)
openscience@royalsociety.org

Appendix A

Dear editor and reviewers:

Thank you very much for giving us an opportunity to revise our manuscript. We appreciate the editor and reviewers very much for the constructive comments and suggestions on our manuscript entitled “A dimensionless parameter coupling the south and north roofs on the thermal environment of Chinese solar greenhouse: Ridge position ratio.”. (ID: RSOS-201707). Those comments are very helpful for revising and improving our paper, as well as the important guiding significance to our research. We have studied the comments carefully and made corrections which we hope meet with approval. The main corrections are marked in red in the manuscript and the responses to the reviewers’ comments are as follows (the replies are highlighted in blue).

Replies to the reviewers’ comments:

Reviewer 1:

Thanks very much for your kind work and consideration on the publication of our paper. On behalf of my co-authors, we would like to express our great appreciation to the editor and reviewers. Our future work will focus on using the model to further optimize the structural design parameters of the Chinese solar greenhouse, such as other important parameters: the ratio of ridge-height to span-length. In addition, a humidity sub-model will be added to this model to further discuss the related problems of high relative humidity when the greenhouse is closed during the night. Such points are as future research plans in the Conclusions. The result concern an empty closed greenhouse is explained in the abstract. Finally, we feel so sorry for our poor English writing ability. Considering the reviewer’s suggestions, we have tried our best to improve paper quality and clarity.

1) Response to comment: The title is too long and possibly confusing. I suggest a simpler one: e.g., Effect of the ridge position on the thermal environment of the Chinese Solar Greenhouse.

Response: The title of this paper has been changed to "Effect of the ridge position ratio on the thermal environment of the Chinese solar greenhouse" according to your suggestion.

2) Response to comment: Symbols are not well represented in the used font, so they must be corrected.

Response: Thank you very much for timely correction, we have corrected the font of the symbol.

3) Response to comment: Ridge Position Ratio (RPR) is defined in the Introduction. The definition should be transferred to the Materials and Methods section.

Response: We have transferred the definition of Ridge Position Ratio (RPR) to the Materials and Methods section.

4) Response to comment: The Materials and Methods section is short, so sub-sections are not necessary.

Response: We have no sub-sections in the Materials and Methods section according to your suggestion.

5) Response to comment: The k- ϵ model and the Boussinesq approximation are well-known models so no detailed explanation is required. A relevant reference could be sufficient.

Response: We have cited relevant references and revised this part of the manuscript.

6) Response to comment: Mesh characteristics could be critical for the quality of the numerical results. For this reason, detailed information regarding the mesh should be provided in the Materials and Methods section.

Response: Thank you very much for your timely reminding, we feel so sorry for our negligence. We have provided detailed information regarding the mesh in the Materials and Methods section.

7) Response to comment: The boundary conditions are not described besides a short statement that “The standard wall function is applied to near-wall processing”. A paragraph is necessary to address in detail the boundary conditions on all areas closing the greenhouse, i.e., north and south roof, soil, north wall, etc.

Response: Thank you very much for your timely reminding, we have addressed in detail the boundary conditions on all areas closing the greenhouse in the manuscript. The specific contents are as follows: “The boundary conditions and initial conditions were determined according to the actual data measured. The inside and outside air temperatures of the greenhouse were 12°C and -20°C initially. The boundary conditions mainly include north and south roofs, soil, north wall setting inside and outside the experimental greenhouse. The south roof was covered with a thin plastic film from 8:30 a.m. to 4:00 p.m. and a heat preservation quilt for the rest of the time. Customized outdoor temperatures are applied to north and south roofs, north wall exterior surfaces through user defined functions (UDF). The inside surface temperatures of north wall, soil, north and south roofs were 17°C, 15°C, 12.5°C and 10°C initially.”.

8) Response to comment: Figure 4: Replace the word interface with cross-section.

Response: We have replaced the word interface with cross-section.

9) Response to comment: Figure 9: Do the fitting curves correspond to equations 10-12?

Response: After our careful check, we have added the corresponding independent variable interval, that is the fitting curves correspond to equations 7-9 (original equations 10-12) when RPR is 0.1-0.45. This makes the conclusion more rigorous.

10) Response to comment: The language needs improvements.

Response: Thank you very much for timely correction, we feel so sorry for our poor English writing ability. We have tried our best to improve paper quality and clarity.

Reviewer 2:

Thanks very much for your kind work and consideration on the publication of our paper. On behalf of my co-authors, we would like to express our great appreciation to the editor and reviewers. Considering the reviewer's suggestions, we tried our best to improve the manuscript. All of the inappropriate clarifications have been modified in the revised manuscript.

1) Response to comment: Literature review: The authors applied a different approach to analyzing the thermal performance of CSG based on the variation of the north roof dimension. However, I found a significant lacking in terms of literature review; several researchers used a similar CFD based model to analyze the thermal performance of CSG. Authors need to state the novelty of their approach to studying thermal performance. The authors also need to state the advantage of their approach compared to the model based on the energy balance equation.

Response: We've listened to your suggestions and revised this part of the manuscript. We have stated the novelty of CFD to studying thermal performance and the advantage of their approach compared to the model based on the energy balance equation according to your suggestion.

2) Response to comment: Unit for Temperature: I see the authors used K as a unit for Temperature, which is not usual for journal publication. I would recommend presenting the results in °C.

Response: We have replaced the unit K with the unit °C for temperature in the manuscript.

3) Response to comment: PRR: I would recommend drawing the line in figure 2 to show the visualization for RPR.

Response: Thank you very much for the advice, we have listened to your suggestion and perfected Figure 2 to visualize RPR.

4) **Response to comment:** Graph: The caption for the graph looks very confusing. Please revise the caption; if the authors like to combine several figures, please present it as a single picture. Figure 4: I don't see any significant difference in temperature variation for different RPR. Authors need to explain the reason for this; if this is the case, then the significance of RPR is not very crucial for the thermal performance of the CSG.

Response: Thank you very much for your timely corrections. We combined several figures and presented them as a single picture. The labeled temperature in Figure 5 is the average air temperature inside the greenhouse. It can be seen from Figure 5a and 5b that the occurrence time of the maximum daytime temperature inside the greenhouse will be delayed from 2 p.m. to 3 p.m. when RPR increases to 0.3. In addition, RPR also has a great effect on the spatial distribution of daytime temperature inside the greenhouse. It can be seen from Figure 5c that RPR has relatively little effect on the minimum temperature at night in each part of the greenhouse. These will be analyzed in detail in the 4.2.1 section. Therefore, the significance of RPR is very crucial for the thermal performance of the CSG.

Once again, thank you very much for your constructive comments and suggestions which would help us both in English and in depth to improve the quality of the paper.

Thank you and best regards.

Yours sincerely,

Name: Tianlai, Li

College of Horticulture, Shenyang Agricultural University, China

E-mail: lxa10157@syau.edu.cn

Appendix B

Dear editor and reviewers:

We appreciate the editor and reviewers very much for the constructive comments and suggestions on our manuscript entitled “Effect of the ridge position ratio on the thermal environment of the Chinese Solar Greenhouse” (ID: RSOS-201707.R1). Those comments are very helpful for revising and improving our paper, as well as the important guiding significance to our research. We have studied the comments carefully and made corrections which we hope meet with approval. The main corrections are marked in red in the manuscript and the responses to the reviewers’ comments are as follows (the replies are highlighted in blue).

Replies to the reviewers’ comments:

Reviewer 1:

Response to comment: I have no further comments on this paper.

Reviewer 2:

1) Response to comment: The article could be accepted with minor revision. Please state the advantages of your model over CSGHEAT and CFD in the introduction section.
Response: As you suggested, we have highlighted the advantages of my model over CSGHEAT and CFD in the introduction section. The statements are marked in red in the revised manuscript.

Once again, thank you very much for your constructive comments and suggestions which would help us in depth to improve the quality of the paper.

Thank you and best regards.

Yours sincerely,

Name: Tianlai Li

College of Horticulture, Shenyang Agricultural University, China

E-mail: lxa10157@syau.edu.cn